# Identification, Typing and Drug Resistance of *Cronobacter* spp. in Powdered Infant Formula and Processing Environment

**DOI:** 10.3390/foods12051084

**Published:** 2023-03-03

**Authors:** Hongxuan Li, Shiqian Fu, Danliangmin Song, Xue Qin, Wei Zhang, Chaoxin Man, Xinyan Yang, Yujun Jiang

**Affiliations:** 1Key Laboratory of Dairy Science, Ministry of Education, Department of Food Science, Northeast Agricultural University, Harbin 150030, China; 2Zhejiang-Malaysia Joint Research Laboratory for Agricultural Product Processing and Nutrition, Key Laboratory of Animal Protein Food Processing Technology of Zhejiang Province, College of Food and Pharmaceutical Sciences, Ningbo University, Ningbo 315800, China

**Keywords:** powdered infant formula, *Cronobacter* spp., antibiotic resistance, multilocus sequence typing, transcriptomics

## Abstract

*Cronobacter* spp. is a food-borne pathogenic microorganism that can cause serious diseases such as meningitis, sepsis, and necrotizing colitis in infants and young children. Powdered infant formula (PIF) is one of the main contamination routes, in which the processing environment is an important source of pollution. In this investigation, 35 *Cronobacter* strains isolated from PIF and its processing environment were identified and typed by 16S rRNA sequencing and multilocus sequence typing (MLST) technology. A total of 35 sequence types were obtained, and three new sequence types were isolated for the first time. The antibiotic resistance was analyzed, showing that all isolates were resistant to erythromycin but sensitive to ciprofloxacin. Multi-drug resistant strains accounted for 68.57% of the total, among which *Cronobacter* strains with the strongest drug resistance reached 13 multiple drug resistance. Combined with transcriptomics, 77 differentially expressed genes related to drug resistance were identified. The metabolic pathways were deeply excavated, and under the stimulation of antibiotic conditions, *Cronobacter* strains can activate the multidrug efflux system by regulating the expression of chemotaxis-related genes, thus, secreting more drug efflux proteins to enhance drug resistance. The study of drug resistance of *Cronobacter* and its mechanism has important public health significance for the rational selection of existing antibacterial drugs, the development of new antibacterial drugs to reduce the occurrence of bacterial resistance, and the control and treatment of infections caused by *Cronobacter*.

## 1. Introduction

*Cronobacter* spp. is a facultatively anaerobic, Gram-negative foodborne pathogen [1]. The World Food and Agriculture Organization (FAO) and the World Health Organization (WHO) have classified *Cronobacter* spp. as one of the pathogenic bacteria in category A of infant formula (PIF). Newborns and infants are high-risk groups for *Cronobacter* infection, and PIF and its processing environment are the main pollution channels of *Cronobacter* spp. [2,3,4]. *Cronobacter* spp. includes seven species and three subspecies [5,6,7]. Among them, *C. sakazakii* and *C. malonaticus* are the main pathogenic bacteria in the clinic. This bacterium is related to meningitis and necrotizing enterocolitis and infects infants by contaminating powdered infant formula (PIF) [8,9], with mortality rates ranging from 40% to 80% [10,11]. Even if cured, there is a possibility of severe neurological sequelae. Multilocus sequence typing (MLST) is a molecular typing method based on DNA sequence. MLST method for *Cronobacter* spp. has become one of the most effective and common methods [12,13]. Currently, the *Cronobacter* MLST database (http://www.pubmlst.org/cronobacter accessed on 22 March 2022) has been established, which contains more than 3000 *Cronobacter* strains from different sources and is divided into more than 800 sequence types (STs). MLST technology has been gradually expanded in the typing research of *Cronobacter* spp., because of its simple operation, high identification, and a high degree of sharing. At present, the study found that the STs of *Cronobacter* have a certain relationship with its serotype [14,15]. In addition, Fei et al. used MLST technology to classify 70 *Cronobacter* strains isolated from PIF in China into 19 STs and found that the dominant STs of *Cronobacter* spp. from PIF in China were ST4, ST1, and ST64 [16]. It can be seen that the MLST method has been widely used in the identification and typing of *Cronobacter* spp. and has been well accepted.

Currently, the most effective treatment for diseases caused by *Cronobacter* spp. is antibiotic therapy [17,18]. Many studies have been carried out to evaluate the drug resistance of *Cronobacter* spp. [19,20,21]. Research reports in recent years have confirmed that *Cronobacter* spp. has strong drug resistance and is increasing year by year [22,23,24]. After research, penicillin, first- and second-generation cephalosporins and other commonly used antibiotics in hospitals have lost their inhibitory effect on *Cronobacter* spp., and new *Cronobacter* isolates with multi-drug resistance spectrum have been gradually discovered [25,26]. Although the multidrug-resistant operon mar was found in the genome of *Cronobacter* spp., the drug resistance level of this genus is generally lower than that of other foodborne pathogens, and the drug resistance mechanism is not yet clear [27,28]. Cao et al. studied the reaction mechanism of *C. sakazakii* CICC 21,544 under the combined stress of citral and carvacrol by transcriptome sequencing and found 25 significantly differentially expressed genes, which were mainly involved in the metabolism of various small molecules, ribosome function, and transmembrane transport systems [29]. Xu et al. used transcriptome sequencing to explore the regulatory mechanism of *pmrA* mutation on the biofilm formation of *C. sakazakii* BAA-894 [30]. The transcriptome research method can be used to determine the known tolerance mechanism. We can also find a new regulatory mechanism related to the tolerance of the bacteria from these differentially expressed genes and reveal the gene function and its possible metabolic pathway. Thereby, contributing to the prevention and control of pathogenic microorganisms in food and its processing environment [31,32,33,34].

Thirty-five *Cronobacter* strains isolated from PIF and its processing environment were used as the research objects in this investigation. The strains were identified and molecularly typed by 16S rRNA sequencing and MLST technology. On this basis, strains with different STs were selected for antibiotic resistance evaluation and analyzed. The drug resistance-related genes and metabolic pathways of *Cronobacter* spp. under antibiotic conditions were mined with the transcriptome analysis, thereby, revealing the resistance mechanism.

## 2. Materials and Methods

### 2.1. Bacterial Strains Information

Thirty-five *Cronobacter* strains isolated from PIF and processing environments in different areas of China were selected in this investigation. The detailed information of strains was listed in Appendix A. Meanwhile, eight type strains (*C. sakazakii* ATCC 29544T, *C. sakazakii* ATCC BAA-894T, *C. malonaticus* LMG 23826T, *C. turicensis* DSM 18703T, *C. muytjensii* ATCC 51329T, *C.universalis* NCTC 9529T, *C. dublinensis* DSM 18705T, *C. condimenti* LMG 26250T) and two reference strains (*C. sakazakii* ATCC 12868, *C. sakazakii* ATCC 29004) were also chosen for comparison.

### 2.2. Reagents and Instruments

A 2× Taq PCR MasterMix and DNA Marker were purchased from TIANGEN Biotech Co., Ltd. (Beijing, China). Primers were synthesized by Sangon Biotech Co., Ltd. (Shanghai, China). EZNATM bacterial total RNA extraction kit was obtained from Omega Bio-Tek Inc. (Norcross, GA, USA). PrimeScriptTM reverse transcription kit was obtained from TaKaRa BIO Co., Ltd. (Dalian, China). Absorbance was measured by SpectraMax i3x multifunctional microplate reader (Molecular Devices Co., Ltd., Sunnyvale, CA, USA). The concentration and purity of the DNA was measured by NanoDrop NC2000 spectrophotometer (Thermo Fisher Scientific Co., Ltd., Shanghai, China).

### 2.3. Strains Culture and DNA Extraction

All strains were cultured at 37 °C, 150 r/min overnight, and then purified on tryptic soy agar (TSA) plates at 37 °C for 14–16 h. A single colony was selected in a tryptic soy broth (TSB) liquid medium and cultured for 8 h at 37 °C 150 r/min to obtain strains at the end of the logarithmic growth phase. In this experiment, TIANGEN bacterial genomic DNA extraction kit was used for rapid extraction of strain DNA and stored at −20 °C for further tests.

### 2.4. 16S rRNA-Based PCR Identification and MLST Technology

The extracted DNA was amplified by PCR using bacterial 16S rRNA universal primers 27-F and 1492-R. The PCR assay was carried out in a 50 μL mixture system including 16 μL of 2× Taq PCR MasterMix, 27 μL of sterile distilled water, 1 μL of each primer with a concentration of 10 μM, and 5 μL of template DNA. The reaction conditions consisted of an initial predenaturation at 95 °C for 5 min, followed by 30 cycles of 94 °C for 30 s, an annealing at 55 °C for 30 s, an elongation at 72 °C for 30 s, and a final extension at 72 °C for 5 min. The amplified products were sequenced to complete the molecular identification of the strains.

At the same time, 7 pairs of housekeeping gene primers of *Cronobacter* spp. were used to specifically amplify the extracted DNA of different strains, the genes were *atpD*, *fusA*, *glnS*, *gltB*, *gyrB*, *infB*, and *pps*, respectively. The amplification system and conditions were the same as the 16S rRNA PCR reaction. Then, the amplified product was sequenced and analyzed, and the unidirectional measurement was performed to obtain the nucleic acid sequences of the 7 housekeeping genes of the strains. The primer sequences were shown in Appendix A.

According to the 16S rRNA sequencing results and the 3036 bp nucleic acid sequence of 7 housekeeping genes spliced in sequence, 35 *Cronobacter* strains were analyzed by neighbor-joining phylogeny analysis using MEGA7.0 software, and the procedure was repeated 1000 times. At the same time, the type strain and reference strain were selected for comparison, and phylogenetic trees based on 16S rRNA and MLST were constructed, respectively.

### 2.5. Drug Resistance Evaluation

For all *Cronobacter* isolates, drug susceptibility tests of 20 commonly used clinical antibiotics in 9 categories were carried out. The operation was performed according to the implementation standard of the American Clinical and Laboratory Standards Institute (CLSI) M100 antimicrobial susceptibility test, and the susceptibility results were interpreted according to its specifications. The names and doses of antibiotics and the judgment results were shown in Table 1.

### 2.6. Transcriptome Sequencing of Representative Drug-Resistant Bacteria

The representative resistant strain was selected for transcriptome sequencing. The strain without antibiotic treatment was activated and cultured under normal conditions. The treatment conditions of the strain requiring antibiotic treatment were as follows: the frozen bacterial solution was removed from −20 °C and added to TSB liquid medium with 2% inoculum for overnight culture, and then reactivated in TSB liquid medium with 4 μg/mL erythromycin for 6 h. The strains before and after treatment were analyzed by transcriptome sequencing. RNA was extracted and purified from the samples and tested for concentration and purity (OD260/280 and OD260/230) along with RNA integrity number (RIN) for quality control (Appendix A). RNA of acceptable quality was subjected to subsequent cDNA synthesis and library construction and then the libraries were double-end sequenced. The raw upland data are filtered to obtain high quality sequences and the filtered sequences are tested for quality using Q20 and Q30 values (Appendix A). The high-quality filtered sequences are only then available for comparison with the reference genome of the species. The expression of each gene was calculated according to the comparison results. On this basis, the expression difference analysis, enrichment analysis, and cluster analysis were carried out.

### 2.7. Verification of Drug Resistance-Related Genes

According to the screening results of transcriptomics tolerance-related genes, 10 genes with a high expression ratio and high correlation with drug resistance were selected for qRT-PCR verification. According to the results of tolerance evaluation, two strains with strong and weak drug resistance were selected for quantitative real-time PCR detection of related genes before and after treatment to investigate the expression differences of each gene in different strains. RNA from strains were extracted with a simple P total RNA extraction kit (Bioer Technology Co., Ltd., Hangzhou, China). NanoDrop spectrophotometer (Thermo, Wilmington, DE, USA) was used to measure the purity and integrity of RNA samples. A reverse transcription kit (Takara Bio, Shiga, Japan) was used for reverse transcription. Additionally, RT-qPCR was performed by SYBR Premix Ex Taq (Takara Bio, Shiga, Japan) according to the manufacturer’s instructions; the reaction was conducted with a QuantStudio™ 3 system. The value of 2^−∆∆Ct^ was calculated to analyze and compare the changes in the expression levels of each gene before and after treatment to verify the accuracy of the omics analysis results. The target genes were matched according to the corresponding sequences in *C. sakazakii* ATCC 29544^T^ genome in the NCBI database, and real-time quantitative primer design was performed in the software Primer Premier 5.0. The gene names and primer sequences were shown in Appendix A.

## 3. Results

### 3.1. 16S rRNA Molecular Identification Results

The 16S rRNA sequencing results of 35 isolates were compared and analyzed in the GenBank database, and the results showed that they all belonged to the genus *Cronobacter* spp., and the similarity between the sequences exceeded 99%, including *C. sakazakii* (*n* = 27, 77.14%), C. malonaticus (*n* = 5, 14.29%), *C. turicensis* (*n* = 2, 5.71%), and *C. dublinensis* (*n* = 1, 2.86%). On this basis, *C. sakazakii* ATCC 29544T, *C. sakazakii* ATCC 12868, *C. malonaticus* LMG 23826T, *C. turicensis* DSM 18703T, and *C. dublinensis* DSM 18705T were used as reference strains, and the phylogenetic tree was constructed according to the 16S rRNA sequencing results, as shown in Figure 1. It can be seen that the strains of the four species of Cronobacter were divided into two large clusters, in which *C. sakazakii* and *C. malonaticus* were divided into one cluster, and *C. turicensis* and *C. dublinensis* were divided into another cluster. The strains of the four species showed clear phylogenetic relationships, on the whole, indicating that 16S rRNA could be used for molecular identification and typing of *Cronobacter* spp.

### 3.2. MLST Analysis of Cronobacter Strains

A total of 35 STs were obtained by MLST typing of *Cronobacter* spp., mainly distributed in 15 homologous complexes. Due to the different sample collection areas and sources, the sequence types were relatively scattered, and some strains could not find the homologous complex matching them in the database, which was the unique type. In addition, three strains of Cronobacter had not been recorded in the database, which were new sequence types discovered for the first time in the world. By uploading to the Cronobacter MLST database (http://www.pubmlst.org/cronobacter accessed on 22 March 2022), the corresponding sequence numbers were obtained, which were one strain of *C. dublinensis* (CD31, ST788) and two strains of *C. sakazakii* (CS32, ST789, and CS33, ST790). The allele numbers of each housekeeping gene and sequence types of all isolates were shown in Table 2.

A phylogenetic tree was constructed based on the nucleic acid sequences of 35 sequence types of 3036 bp and 10 strains of Cronobacter were used as a reference to evaluate the genetic relationship among different sequence types of strains. The results were shown in Figure 2. All strains were divided into five large clusters and seven species kept a certain distance from each other, among which *C. sakazakii* and *C. malonaticus* showed a closer relationship. Within *C. sakazakii* species, ST4 and ST268, ST13 and ST789, ST64 and ST261, ST73, ST269 and ST790 had closer phylogenetic relationships. Among C. malonaticus species, ST7 and ST201 were closely related in phylogeny. According to the results of the MLST database, there was only one base difference between these similar sequence types. Therefore, the strains corresponding to each sequence type were likely to have similar phylogenetic relationships at the whole genome level.

### 3.3. Drug Resistance Analysis

The drug susceptibility results of 35 Cronobacter strains were shown in Table 3. All Cronobacter isolates had the highest resistance rate to erythromycin (100%), followed by sulfamethoxazole/trimethoprim (45.71%). The others were neomycin (37.14%), cefazolin (28.57%), kanamycin (28.57%), ceftriaxone (25.71%), amikacin (25.71%), ampicillin (20%), ceftazidime (20%), cefuroxime (14.29%), polymyxin B (14.29%), gentamicin (11.43%), norfloxacin (11.43%), piperacillin (8.57%), cefoperazone (8.57%), tetracycline (8.57%), doxycycline (8.57%), ofloxacin (2.86%), and chloramphenicol (2.86%). All strains were sensitive to ciprofloxacin.

The drug resistance profile of Cronobacter strains was shown in Table 4. There were six strains with one drug resistance spectrum (erythromycin), accounting for 17.14%. The double drug resistance spectrum showed three spectrum types and a total of five strains, accounting for 14.29%. There were six types of triple-drug resistance spectrum and seven strains in total, accounting for 20%. The quadruple drug resistance spectrum includes four spectrum types and a total of four strains, accounting for 11.43%. The five-fold drug resistance spectrum showed three spectrum types and a total of three strains, accounting for 8.57%. The six-fold drug resistance spectrum showed two spectrum types and a total of two strains, accounting for 5.71%. The seven-fold drug resistance spectrum showed six spectrum types and a total of six strains, accounting for 17.14%. There was one strain in each of the eleven and thirteen-fold drug resistance spectrums, accounting for 2.86%. According to the results, 24 strains were multidrug resistant (resistant to three or more classes of antibiotics), accounting for 68.57% of the total number of Cronobacter isolates. At the same time, *C. sakazakii* ATCC 29544 was only resistant to erythromycin and sulfamethoxazole/trimethoprim, which was a double drug resistance. In conclusion, compared with the reference strains, most Cronobacter isolates showed strong drug resistance and multi-drug resistance, among which the representative drug-resistant strains were CS14 (ST42) and CS17 (ST64).

### 3.4. Transcriptome Analysis of Strains under Antibiotic Conditions

Based on the results of the drug resistance test, CS14, which has a high ability to survive under antibiotic conditions, was selected as the representative strain. The normal culture group (A) and the antibiotic treatment group (B) were compared with each other, and the overall situation of the screened differentially expressed genes is shown in Figure 3. Figure 3a as a volcano map of differentially expressed genes. Translated with www.DeepL.com/Translator (free version, accessed on 3 April 2022), a total of 2324 differentially expressed genes were screened, among which 1154 genes were upregulated and 1170 genes were downregulated. The cluster heat map of differentially expressed genes was obtained by two-way cluster analysis on the union of screened differentially expressed genes and samples. As can be seen in Figure 3b, the gene expression between the three duplicate samples of the control group and the treatment group was consistent, and the gene expression between the comparison groups was significantly different.

KEGG enrichment analysis was performed on differentially expressed genes and the top 30 pathways with the most significant enrichment were selected for demonstration. The results were shown in Figure 4. Signaling pathways can be divided into three categories: environmental information processing, genetic information processing, and metabolism. The metabolic category contained 26 signaling pathways and 381 differentially expressed genes, which mainly focused on carbohydrate synthesis and metabolism, energy metabolism, amino acid metabolism, terpenoids and polyketides metabolism, lipid metabolism, degradation of heterogeneous organisms, and metabolic processes. The genetic information processing category included three signaling pathways and 75 differentially expressed genes, mainly focusing on ribosomal translation and RNA transcription signaling pathways. The environmental information processing category included one signaling pathway and 115 differentially expressed genes, focusing on membrane transport processes.

According to the results of RNA-seq analysis under antibiotic treatment, a total of 94 genes related to drug resistance were screened, among which 77 genes were differentially expressed, 35 genes were upregulated, and 42 genes were downregulated. Some differentially expressed genes were screened in Table 5, listed in order of *p* value from smallest to largest.

### 3.5. qRT-PCR Validation of Drug Resistance-Related Genes

Ten differentially expressed genes related to drug resistance were screened and qRT-PCR was performed on CS14 and CS17 strains with multiple drug resistance, and CS6 and CS9 strains with only one double drug resistance. The results were shown in Table 6. The relative expression of CS14 genes was consistent with the transcriptome analysis results. Eight genes were upregulated and significantly upregulated (2^−ΔΔCt^ ≥ 2, *p* < 0.05), and two genes were downregulated and significantly downregulated (2^−ΔΔCt^ ≤ 0.5, *p* < 0.05). The relative gene expression of CS17 was consistent with that of CS14, among which six genes were significantly upregulated and two genes were significantly downregulated. For CS6 with weak drug resistance, eight out of ten genes were downregulated, among which four genes were significantly downregulated, and no genes were significantly upregulated. For another strain CS9 with weak drug resistance, seven out of ten genes were downregulated, including two significantly downregulated genes, and no genes that were significantly upregulated. In general, the results of qRT-PCR were consistent with the transcriptome, with higher upregulation or downregulation levels in the two resistant strains compared with the two less resistant strains.

### 3.6. Analysis of Drug Resistance Mechanism

When the strain was challenged with antibiotics in the bacterial chemotaxis pathway (ko02030), the genes CSK29544_01507 and CSK29544_02725 encoding methyl receptor chemotactic proteins were significantly upregulated by 2.24 and 2.66-fold, respectively. The gene CSK29544_02809 encoding the methyl receptor chemotactic citrate sensor was significantly upregulated by 2.36-fold; the gene CSK29544_00467 encoding the methyl receptor chemosensory sensor was significantly upregulated by 2.06 times; and the genes *fliG* and *fliM* encoding the flagellar motor switch protein were significantly upregulated by 1.76 and 1.83 times, respectively. At the same time, the genes CSK29544_01807 and CSK29544_02659 encoding methyl receptor chemosensory sensors were significantly downregulated by 1.57 and 2.21 times, respectively. The gene *cheR* encoding chemotactic protein methyltransferase was significantly downregulated by 1.79 times. The gene *cheB* encoding the chemotactic family, protein-glutamate methyl esterase/glutaminase, was significantly down-regulated by 2.51 times. The gene encoding purine-binding chemotactic protein *cheW* was significantly downregulated by 2.20 times; and the gene encoding chemotactic protein *cheZ* was significantly downregulated by 2.22 times. It can be seen that Cronobacter can enhance the drug resistance of bacteria as a whole by regulating the partial upregulation or downregulation of chemotaxis-related genes.

In the ABC transporter pathway (KO02010), 29 genes were significantly upregulated, and 86 genes were significantly downregulated. The gene *mdlA* encoding the ATP-binding protein of the multidrug efflux system was significantly upregulated by 2.05 times, and the gene *yojI* encoding the ATP-binding/permease protein of the multi-drug transport system was significantly upregulated by 1.96 times. The four coding genes *potA*, *potB*, *potC*, and *potD* involved in spermidine transport were significantly upregulated by 2.46, 3.10, 2.5,9 and 2.06 times, respectively. The gene *proX* encoding the glycine betaine/proline transport system substrate binding protein was significantly upregulated by 2.76 times; the gene *proW* encoding the glycine betaine/proline transport system permease protein was significantly upregulated by 2.60 times; and encoding the glycine betaine/proline gene *proV* of the acid transport system ATP-binding protein was significantly upregulated 4.23-fold. The genes *togM* and *togN* encoding the oligogalacturonic acid transport system permease proteins were significantly upregulated by 2.11 and 2.37 times, respectively, and the gene *togB* encoding the low-galacturonic acid transport system substrate-binding protein was significantly downregulated by 2.39 times. The three encoding genes *mlaD*, *mlaE*, and *mlaF* involved in phospholipid transport were significantly upregulated by 1.88, 2.63, and 1.88 times, respectively. The genes encoding glutamine transport *glnP* and *glnQ* were significantly upregulated; the genes involved in zinc ion transport *znuB* and *znuC* were significantly upregulated; the genes involved in lipoprotein transport *lolC*, *lolD*, and *lolE* were significantly upregulated; and the encoding genes *lptF* involved in lipopolysaccharide transport were significantly upregulated and *lptG* were significantly upregulated.

In the two-component signaling pathway (ko02020), 25 genes were significantly upregulated and 46 genes were significantly downregulated. The gene *baeR* encoding the *ompR* family and response regulator was significantly downregulated by 2.20 times. The gene *mdtA* encoding the multidrug efflux system membrane fusion protein was significantly downregulated by 1.50 times. The gene *mdtD* encoding MFS transporter, DHA2 family, multidrug resistance protein was significantly downregulated by 2.26 times. The gene *acrD* encoding the MDR efflux system was significantly downregulated by 2.01-fold. In summary, under the stimulation of antibiotics, *Cronobacter* spp. secretes more drug efflux proteins, enhances drug resistance by activating the multidrug efflux system, and reduces the transport activity of nucleic acids, lipids, amino acids, and other molecular substances. The damaging effect of drugs on cells.

## 4. Discussion

In this study, 35 STs were obtained by MSLT on all isolated *Cronobacter* spp. strains, which were distributed in raw materials, semi-finished products, finished products, and the processing environment. The main sequence types were ST1 (14.42%), ST4 (18.27%), and ST64 (11.54%). This study found that ST4 was present in the finished product, and many studies have reported that ST4 was associated with infantile meningitis [35]. ST1 has also been isolated from clinically infected infants [36]. ST64 is one of the main contamination sequence types of PIF and its processing environment in China, enough to show its drug resistance and difficulty to disinfect [16]. Therefore, it is very important to study different sequence types of *Cronobacter* spp. to reduce the contamination of *Cronobacter* spp. in PIF. In addition, three of the isolates were not recorded in the database and were new sequence types found for the first time in the world. Phylogenetic trees were constructed based on the MLST results and one representative strain for each ST was selected for follow-up studies.

The isolates showed 100% resistance to erythromycin, followed by 45.71% (16/35) to sulfamethoxazole/trimethoprim, indicating that *Cronobacter* spp. used in this study had a good tolerance rate to macrolides and sulfonamides antibiotics. Meanwhile, all tested isolates were sensitive to ciprofloxacin. We found that the strains were only 11.43% and 2.86% resistant to norfloxacin and ofloxacin (both belong to fluoroquinolones), so the contamination of *Cronobacter* spp. could be treated with fluoroquinolones. This study also found that strains with only one drug resistance spectrum (erythromycin) accounted for 17.14% (6/35), and multidrug-resistant strains accounted for 68.57% (24/35), indicating that the overall drug resistance of *Cronobacter* spp. was strong. Among them, CS14 (ST42) and CS17 (ST64) were the most representative drug-resistant strains, which were 13 and 11 multiple drug resistance, respectively.

In recent years, many reports have found that a variety of antibiotics, such as gentamicin, ampicillin, kanamycin, and ciprofloxacin, can kill *Cronobacter* spp. [37]. Lai et al. found that *Cronobacter* spp. had consistent resistance to ampicillin, cefazolin, and extended-spectrum penicillin [38]. Kim et al. found that *Cronobacter* spp. isolated from food was resistant to ceftaroline and ampicillin [39]. It was further confirmed that even though antibiotics can be effective in eliminating the contamination caused by this bacteria, long-term use can cause multi-drug resistance in *Cronobacter* spp. Pakbin et al. showed that *C. sakazakii* isolates were completely resistant to ampicillin and amoxicillin and showed multidrug resistance, and moderate resistance to ciprofloxacin and tetracycline antibiotics [40]. Odeyemi et al. found that *Cronobacter* spp. was resistant to erythromycin and sulfamethoxazole, and all bacteria were able to form biofilms [41]. This finding is highly consistent with the findings of this experiment. In conclusion, multi-drug resistance of *Cronobacter* spp. is increasing year by year, and screening for effective antibiotics is one of the urgent research priorities.

In this study, a total of 77 differentially expressed genes related to drug resistance were screened by transcriptome analysis of CS14 under antibiotic conditions, and some highly correlated genes included *emrA*, CSK29544_03309, *yojI*, *mdfA*, *fliM*, *mdlA*, *emrB*, *pump*, CSK29544_03853, and *baeR*. We found that the regulatory genes related to methyl-accepting chemotaxis were significantly upregulated by about two-fold, and the regulatory genes fliG and fliM related to flagellar motor switch protein were significantly upregulated by 1.76 and 1.83-fold, respectively. In addition, the regulatory genes *mdlA* and *yojI* related to the multidrug efflux system were also significantly upregulated by approximately two-fold, as were some genes involved in amino acid, sugar, and lipid transport. In summary, under the stimulation of antibiotic conditions, *Cronobacter* spp. can activate the multidrug efflux system by regulating the expression of chemotaxis-related genes, thereby, secreting more drug efflux proteins to enhance drug resistance. The results of the study also revealed that *Cronobacter* spp. can reduce cellular damage from drugs by reducing the transport activity of nucleic acids, lipids, amino acid, and other molecular substances. Some researchers found the multi-drug resistance operon *mar* in the genome of *Cronobacter* spp. Bao et al. also found that the polymyxin resistance gene *pmrA* was related to biofilm formation in *Cronobacter* spp., but, overall, the antibiotic resistance level of this genus was lower than that of other foodborne pathogens [42]. However, the results of this study showed that the *Cronobacter* strains had a strong drug resistance level and a multi-drug resistance spectrum, which may be due to the stronger resistance of the *Cronobacter* spp. from PIF and its processing environment.

## 5. Conclusions

In conclusion, the 35 *Cronobacter* strains isolated from PIF and its processing environment showed high diversity. A total of 35 sequence types were obtained by MLST typing, among which three sequence types were discovered for the first time in the world: *C. dublinensis* CD31 (ST788), *C. sakazakii* CS32 (ST789), and CS33 (ST790), respectively. The resistance rate of all isolates to erythromycin was 100%, but they were all sensitive to ciprofloxacin, and the multidrug-resistant strains accounted for 68.57%, among which the most resistant strain reached 13 multiple drug resistance. Seventy-seven differentially expressed genes related to drug resistance were screened by transcriptome sequencing and the qRT-PCR verification results were consistent with the transcriptome results. Under the stimulation of antibiotic conditions, *Cronobacter* strains can activate the multidrug efflux system by regulating the expression of chemotaxis-related genes, thus, secreting more drug efflux proteins to enhance drug resistance. These studies can provide a theoretical basis for the prevention and treatment of *Cronobacter*.

## Figures and Tables

**Figure 1 foods-12-01084-f001:**
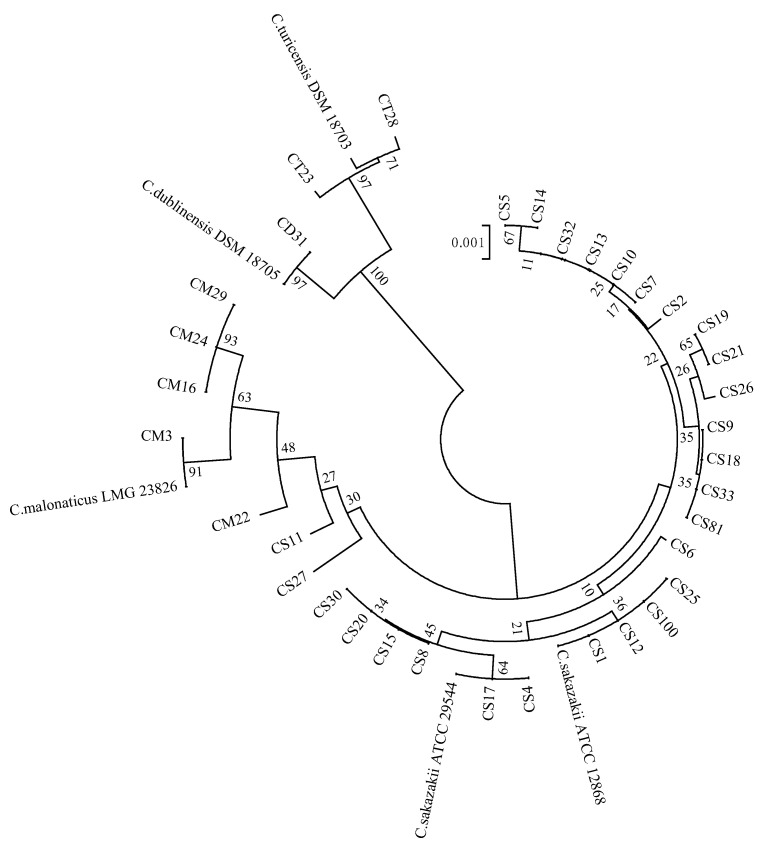
The phylogenetic tree based on 16S rRNA sequences.

**Figure 2 foods-12-01084-f002:**
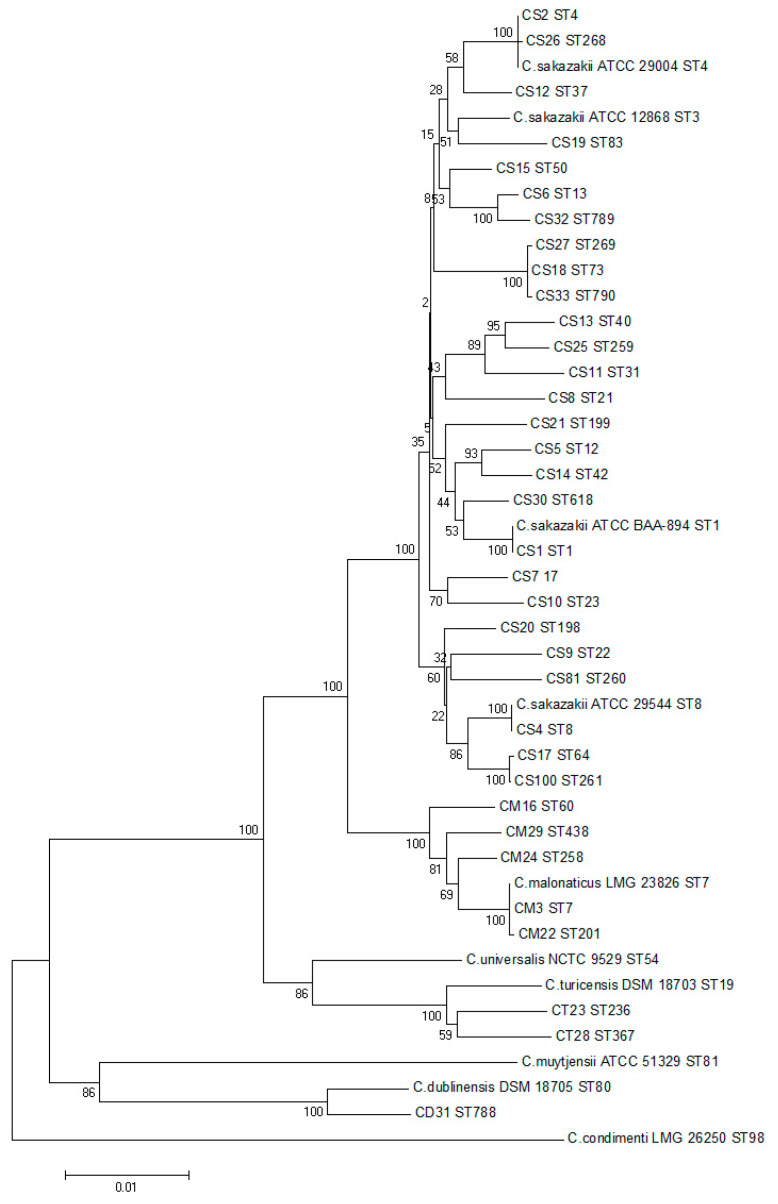
Phylogenetic tree based on the different ST type housekeeping genes.

**Figure 3 foods-12-01084-f003:**
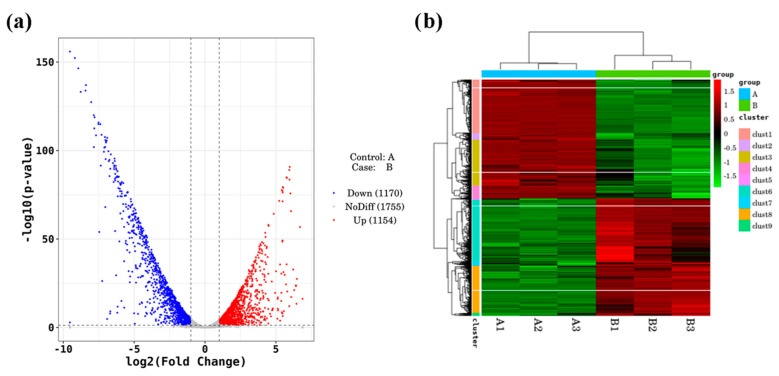
Volcano map and cluster heat map of differentially expressed genes. (**a**) Volcano map; (**b**) cluster heat map.

**Figure 4 foods-12-01084-f004:**
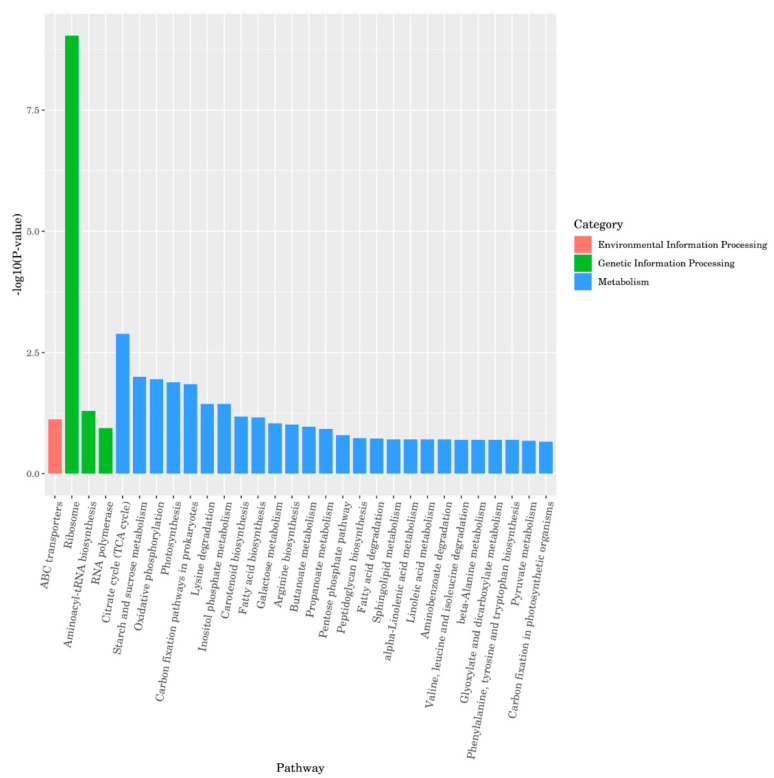
KEGG enrichment analysis of CS14.

**Table 1 foods-12-01084-t001:** Antibiotics for drug sensitivity determination and judgment criteria.

Antimicrobial Group	Antibiotics	Content (μg/tablet)	Diameter of Inhibition Zone (mm)		
			Resistance	Intermediary	Sensitivity
penicillins					
1	ampicillin	10	≤13	14–16	≥17
2	piperacillin	100	≤17	18–20	≥21
cephalosporins					
3	cefazolin	30	≤14	15–17	≥18
4	cefuroxime	30	≤14	15–17	≥18
5	ceftazidime	30	≤17	18–20	≥21
6	ceftriaxone	30	≤19	20–22	≥23
7	cefoperazone	75	≤15	16–20	≥21
aminoglycosides					
8	amikacin	30	≤14	15–16	≥17
9	gentamicin	10	≤12	13–14	≥15
10	kanamycin	30	≤13	14–17	≥18
11	neomycin	30	≤12	13–16	≥17
tetracyclines					
12	tetracycline	30	≤11	12–14	≥15
13	doxycycline	30	≤10	11–13	≥14
macrolides					
14	erythromycin	15	≤13	14–22	≥23
fluoroquinolones					
15	norfloxacin	10	≤12	13–16	≥17
16	ofloxacin	5	≤12	13–15	≥16
17	ciprofloxacin	5	≤15	16–20	≥21
polypeptides					
18	polymyxin B	300	≤8	8–11	≥12
sulfonamides					
19	sulfamethoxazole/trimethoprim	23.75/1.25	≤10	11–15	≥16
Phenylpropanols					
20	chloramphenicol	30	≤12	13–17	≥18

**Table 2 foods-12-01084-t002:** MLST typing results of *Cronobacter* strains.

Strains	*atpD*	*fusA*	*glnS*	*gltB*	*gyrB*	*infB*	*ppsA*	ST	CC
CS1	1	1	1	1	1	1	1	1	1
CS2	5	1	3	3	5	5	4	4	4
CM3	10	7	6	7	9	14	9	7	7
CS4	11	8	7	5	8	15	10	8	8
CS5	18	17	10	12	18	24	18	12	-
CS6	15	14	15	13	22	5	16	13	13
CS7	3	12	16	5	16	20	14	17	17
CS8	3	11	13	18	11	17	13	21	21
CS9	16	1	19	19	26	5	26	22	-
CS10	20	18	16	10	3	20	27	23	23
CS11	3	8	37	22	29	36	32	31	31
CS12	44	15	3	5	5	38	59	37	37
CS13	3	15	28	22	5	38	19	40	40
CS14	48	17	10	69	71	5	81	42	-
CS15	3	8	13	15	22	20	21	50	-
CM16	12	7	8	8	10	16	43	60	-
CS17	16	8	13	40	15	15	10	64	64
CS18	55	14	59	70	70	70	80	73	73
CS19	19	16	19	41	19	15	23	83	83
CS20	3	8	3	3	18	46	127	198	52
CS21	15	8	13	94	99	98	126	199	-
CM22	10	7	6	99	9	14	9	201	7
CT23	47	39	101	121	118	49	153	236	-
CM24	89	13	107	8	10	35	160	258	-
CS25	16	37	108	22	74	56	4	259	-
CS81	16	8	9	109	125	3	102	260	-
CS100	16	8	13	40	15	15	161	261	64
CS26	5	1	3	127	5	5	4	268	4
CS27	55	14	59	128	70	70	80	269	73
CT28	115	5	144	165	158	29	203	367	-
CM29	117	7	40	181	188	165	120	438	-
CS30	3	18	9	267	1	225	210	618	-
CD31	95	23	114	123	157	132	171	788	-
CS32	15	14	15	13	22	5	347	789	-
CS33	55	14	59	70	70	70	389	790	-

Note: “-” means that the clonal complex (CC) corresponding to the strain does not exist in the database.

**Table 3 foods-12-01084-t003:** Single antibiotic resistance of 35 Cronobacter strains.

Antibiotics	Resistance Rate (%)	Mediation Rate (%)	Sensitivity Rate (%)
ampicillin	20.00	5.71	74.29
piperacillin	8.57	11.43	80.00
cefazolin	28.57	40.00	31.43
cefuroxime	14.29	5.71	80.00
ceftazidime	20.00	5.71	74.29
ceftriaxone	25.71	14.29	60.00
cefoperazone	8.57	45.71	45.71
amikacin	25.71	20.00	54.29
gentamicin	11.43	20.00	68.57
kanamycin	28.57	42.86	28.57
neomycin	37.14	57.14	5.71
tetracycline	8.57	5.71	85.71
doxycycline	8.57	0	91.43
erythromycin	100	0	0
norfloxacin	11.43	17.14	71.43
ofloxacin	2.86	5.71	91.43
ciprofloxacin	0	0	100
polymyxin B	14.29	20.00	65.71
sulfamethoxazole/trimethoprim	45.71	22.86	31.43
chloramphenicol	2.86	14.29	82.86

**Table 4 foods-12-01084-t004:** Antibiotic resistance profile analysis of Cronobacter isolates.

Multiple Numbers	Drug Resistant Spectrum	Strain Number	Total
1	erythromycin	6, 9, 11, 18, 26, 100	6 (17.14%)
2	amikacin-erythromycin	7	5 (14.29%)
	cefazolin-erythromycin	8, 16, 19	
	kanamycin-erythromycin	21	
3	gentamicin-doxycycline-erythromycin	2	7 (20.00%)
	ampicillin-cefazolin-erythromycin	4, 81	
	gentamicin-sulfamethoxazole/trimethoprim-erythromycin	12	
	cefazolin-sulfamethoxazole/trimethoprim-erythromycin	13	
	neomycin-sulfamethoxazole/trimethoprim-erythromycin	15	
	ceftriaxone-amikacin-erythromycin	32	
4	ampicillin-cefoperazone-neomycin-erythromycin	3	4 (11.43%)
	cefuroxime-tetracycline-polymyxin B-erythromycin	5	
	neomycin-tetracycline-sulfamethoxazole/trimethoprim-erythromycin	28	
	ceftazidime-ceftriaxone-sulfamethoxazole/trimethoprim-erythromycin	33	
5	kanamycin-neomycin-polymyxin B-chloramphenicol- erythromycin	10	3 (8.57%)
	amikacin-kanamycin-norfloxacin-sulfamethoxazole/trimethoprim-erythromycin	22	
	cefazolin-amikacin-kanamycin-norfloxacin-erythromycin	30	
6	cefazolin-amikacin-kanamycin-neomycin-sulfamethoxazole/trimethoprim-erythromycin	20	2 (5.71%)
	cefazolin-amikacin-kanamycin-neomycin-sulfamethoxazole/trimethoprim-erythromycin	31	
7	piperacillin-amikacin-kanamycin-neomycin-doxycycline-sulfamethoxazole/trimethoprim-erythromycin	1	6 (17.14%)
	ceftazidime-cefazolin-amikacin-neomycin-norfloxacin-sulfamethoxazole/trimethoprim-erythromycin	23	
	cefazolin-amikacin-gentamicin-kanamycin-neomycin-sulfamethoxazole/trimethoprim-erythromycin	24	
	ceftazidime-gentamicin-kanamycin-neomycin-norfloxacin-sulfamethoxazole/trimethoprim-erythromycin	25	
	ampicillin-cefuroxime-ceftazidime-cefazolin-polymyxin B-sulfamethoxazole/trimethoprim-erythromycin	27	
	ampicillin-cefuroxime-ceftazidime-cefazolin-neomycin-sulfamethoxazole/trimethoprim-erythromycin	29	
11	ampicillin-piperacillin-cefazolin-cefuroxime-ceftazidime-cefazolin-cefoperazone-neomycin-polymyxin B-sulfamethoxazole/trimethoprim-erythromycin	17	1 (2.86%)
13	ampicillin-piperacillin-cefazolin-cefuroxime-ceftazidime-cefazolin-cefoperazone-neomycin-tetracycline-ofloxacin-polymyxin B-sulfamethoxazole/trimethoprim-erythromycin	14	1 (2.86%)

**Table 5 foods-12-01084-t005:** Drug resistance-related differentially expressed genes.

Gene	Gene Annotation	log_2_FoldChange	*p*-Value
CSK29544_00553	permease of the drug/metabolite transporter (DMT) superfamily	3.77	2.65 × 10^−43^
CSK29544_03853	putative ABC transporter, ATP-binding protein	−4.46	3.53 × 10^−43^
*emrA* (CSK29544_01824)	multidrug efflux system protein	3.27	1.39 × 10^−29^
CSK29544_03309	multidrug efflux protein	2.81	3.63 × 10^−25^
CSK29544_02809	methyl-accepting chemotaxis citrate transducer	2.36	7.87 × 10^−16^
CSK29544_00467	methyl-accepting chemotaxis sensory transducer	2.06	1.74 × 10^−13^
*yojI* (CSK29544_02362)	multidrug transport system ATP-binding/permease protein	1.96	4.79 × 10^−13^
*baeR* (CSK29544_02460)	two-component system, OmpR family, response regulator	−2.20	1.27 × 10^−12^
CSK29544_01823	multidrug resistance protein B	1.97	1.45 × 10^−10^
*mdfA* (CSK29544_03801)	multidrug efflux system translocase	2.41	2.35 × 10^−8^
*mdtA* (CSK29544_02465)	membrane fusion protein, multidrug efflux system	−1.50	8.72 × 10^−7^
*fliM* (CSK29544_02582)	flagellar motor switch protein	1.83	1.66 × 10^−6^
*eamA* (CSK29544_03092)	O-acetylserine/cystein export protein	−1.82	4.17 × 10^−6^
*mdlA* (CSK29544_04123)	ATP-binding cassette, multidrug efflux pump	2.05	2.38 × 10^−5^
CSK29544_01507	methyl-accepting chemotaxis protein	2.24	3.26 × 10^−5^
*mdtD* (CSK29544_02462)	MFS transporter, DHA2 family, multidrug resistance protein	−2.26	1.53 × 10^−4^
*emrB* (CSK29544_03823)	EmrB subfamily drug resistance transporter	2.13	4.00 × 10^−4^
*emrE* (CSK29544_03219)	multidrug transporter	−2.18	5.47 × 10^−4^
*acrD* (CSK29544_01993)	multidrug efflux pump	−2.01	1.04 × 10^−3^
CSK29544_00887	multidrug resistance protein D	2.48	1.50 × 10^−2^
*pump* (CSK29544_00631)	transcription repressor of multidrug efflux	2.50	2.41 × 10^−2^

**Table 6 foods-12-01084-t006:** Changes in relative expression levels of some drug resistant differentially expressed genes by qRT-PCR.

Gene	Relative Transcript Level (2^−ΔΔCt^)			
	CS6	CS9	CS17	CS14
*emrA* (CSK29544_01824)	1.25 ± 0.35	1.31 ± 0.28	6.85 ± 0.26	7.14 ± 0.21
CSK29544_03309	0.94 ± 0.22	1.27 ± 0.40	5.09 ± 0.15	5.13 ± 0.34
*yojI* (CSK29544_02362)	0.36 ± 0.12	0.46 ± 0.17	1.87 ± 0.19	2.20 ± 0.17
*mdfA* (CSK29544_03801)	0.44 ± 0.20	0.83 ± 0.21	4.98 ± 0.47	5.06 ± 0.28
*fliM* (CSK29544_02582)	0.37 ± 0.09	0.65 ± 0.25	1.96 ± 0.23	2.31 ± 0.23
*mdlA* (CSK29544_04123)	0.99 ± 0.29	1.02 ± 0.22	2.25 ± 0.28	2.17 ± 0.22
*emrB* (CSK29544_03823)	0.71 ± 0.43	0.58 ± 0.17	3.39 ± 0.26	3.22 ± 0.41
*pump* (CSK29544_00631)	0.66 ± 0.21	0.43 ± 0.14	5.53 ± 0.37	5.66 ± 0.20
CSK29544_03853	1.34 ± 0.25	0.85 ± 0.23	0.38 ± 0.06	0.32 ± 0.16
*baeR* (CSK29544_02460)	0.45 ± 0.11	0.79 ± 0.22	0.46 ± 0.17	0.47 ± 0.10

## Data Availability

Data in the project are still being collected, but all data used in the study are available by contacting the authors.

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
