# Peer review of "Identification, Typing and Drug Resistance of Cronobacter spp. in Powdered Infant Formula and Processing Environment"

_foods, 2023, doi:10.3390/foods12051084_

Round 1

Reviewer 1 Report

Li et al. in the presented paper assessed the diversity of 35 Cronobacter sp. strains using MLST. Also, resistance to 20 antibiotics was evaluated. Some parts of the manuscript are not clear. What was the aim of RNAseq? I guess the authors tried to study the mechanism underlying AMR. Why was the transcriptome of only 1 strain analyzed? Why was erythromycin chosen? To study the mechanism the authors should compare more transcriptomes of both sensitive and drug-resistant strains exposed and not exposed to selected antibiotics. How RNA for transcriptome analysis and qPCR was isolated? How its quality was evaluated? The procedure for RNAseq is not included. Also, qPCR procedure is missing. What was the reference gene used for expression evaluation?

Reviewer 2 Report

The main question addressed by this research refers to food safety, the isolation and identification of Cronobater strains and the analysis of antibiotic resistance of these strains.

It is a relevant and interesting study because the presence of pathogenic germs resistant to antibiotics in food (powdered infant formula) can cause serious diseases with an impact on children's health.

Although many studies have been carried out to evaluate the drug resistance of Cronobacter spp., the authors chose to study the 35 Cronobacter strains isolated from powdered infant formula and its processing environment from the point of view of antibiotic resistance evaluation. The authors demonstrated that Cronobacter strains can activate the multidrug efflux system by regulating the expression of chemotaxis-related genes, thus secreting more drug efflux proteins to enhance drug resistance.

The authors state that they obtained a total of 35 sequence types, among which 3 sequence types were discovered for the first time in the world: C. dublinensis CD31 (ST788), C. sakazakii CS32 (ST789) and CS33 (ST790. This research can provide a theoretical basis for the prevention and treatment of Cronobacter.

I consider the paper is well written. The text is clear and easy to read. The Materials and Methods sections as well as the Results are presented clearly and in detail. The conclusions are consistent with the evidence and arguments presented.

The manuscript presents a complex and interesting study about the identification and drug resistance of Cronobacter spp. in baby food. The authors state that they obtained a total of 35 sequence types by MLST typing, among which 3 sequence types were discovered for the first time in the world. The resistance rate of all isolates against several antibiotics was studied, and a relationship between antibiotic conditions and the multidrug efflux system in Cronobacter strains was shown.

Some observations (regarding the form of the text)

1. Line 71 "we" - We

2. Line 117 – Cronobacter in italics

3. In table 4 – multiple numbers 3 – for cefazolin-gentamicin-sulfamethoxazole/trimethoprim-erythromycin- in this case there are 4 antibiotics

4. For figures 3 and 4 – the title of the figure should be written below the figure

5. Table 5 "emrA (CSK29544_01824) ultidrug efflux system protein" - multidrug

Reviewer 3 Report

General:

The research is important in food safety and the manuscript is well presented. Just few points to consider:

Abstract:

The abstract needs a revision:

1.    Put a clear statement about the objective (s) of the research and the practical application of the research as conclusions

Did you collect the Cronobacter strains from PIF or get the strains  from the labs?

In the Key words, better to use "antibiotic resistance" than drug resistance

Did you see any inter-Cronobacter strains (Eight type strains mentioned in the manuscript) variation for antibiotic resistance

In a Material and Methods mention the methods used  to determine the level of antibiotic resistance. (I assumed you use  Kirby Bauer disk diffusion method)

 Advisable to mention which Cronobacter strains were from powdered infant formula or processing environment

May be helpful if  the  correlations between antibiotic resistance  and the Cronobacter strains is checked from the original data

Round 2

Reviewer 1 Report

Since Cronobacter is a procaryota the good  quality  and integrity of RNA should be confirmed by 16S and 23S bands and not 28S and 18S. There is also no information about the treatment with DNAse.

Round 3

Reviewer 1 Report

Thank you for the proof.